# The relationship of mRNA with protein expression in CD8+ T cells associates with gene class and gene characteristics

**Benoît P. Nicolet**[1,2,3¤], **Monika C. Wolkers**[1,2,3]*

1 Department of Hematopoiesis, Sanquin Research, Amsterdam, The Netherlands, 2 Landsteiner Laboratory, Amsterdam UMC, University of Amsterdam, Amsterdam, The Netherlands, 3 Oncode Institute, Utrecht, The Netherlands

¤ Current address: Division of Molecular Oncology & Immunology, The Netherlands Cancer Institute, Amsterdam, The Netherlands
* m.wolkers@sanquin.nl

## Abstract

T cells are key players in our defence against infections and malignancies. When T cells differentiate or become activated, they undergo substantial alterations in gene expression. Even though RNA expression levels are now well documented throughout different stages of T cells, it is not well understood how mRNA expression translates into the protein landscape. By combining paired RNA sequencing and mass spectrometry data of primary human CD8+ T cells, we report that mRNA expression is a poor proxy for the overall protein output, irrespective of the differentiation or activation status. Yet, gene class stratification revealed a function-specific correlation of mRNA with protein expression. This gene class-specific expression pattern associated with differences in gene characteristics such as sequence conservation and untranslated region (UTR) lengths. In addition, the presence of AU-rich elements in the 3'UTR associated with alterations in mRNA and protein abundance T cell activation dependent, gene class-specific manner. In conclusion, our study highlights the role of gene characteristics as a determinant for gene expression in T cells.

## Introduction

CD8+ T cell responses are key players in clearing and controlling microbial infections and tumor cells. To exert their effector function, naïve T cells become activated upon antigen recognition, and they differentiate into effector T cells (reviewed in [1]). T cells also mature into different subsets of memory T cells, which often provides us with life-long protection from severe sickness by recurring infections [2, 3]. This protection is achieved by their capacity to become rapidly reactivated upon antigen recognition, resulting in the rapid production of effector molecules, such as granzymes, chemokines, and cytokines [1, 4–6].

T cell differentiation and activation depends on substantial alterations of the transcriptional landscape, and on metabolic reprogramming [7–9]. In CD8+ T cells, dynamic genome-wide alterations were observed during infections in mice and humans at the transcript levels [7, 10]. However, data integration of the transcriptome and proteome revealed a striking discrepancy

**Data Availability Statement:** All data sets are publicly available from GEO (NCBI) at accession numbers GSE63144 and GSE125497 and from PRIDE (EBL) at DOI:10.1073/pnas.1913940117 (PXD012874). All script are available from Github

at https://github.com/BenNicolet/mRNA_and_protein_intregration_in_CD8_Tcells.

**Funding:** Our study was supported by the Oncode Institute (grant to M. Wolkers), and the European research council (ERC-Printers 817533). These funding bodies had no role in study design, data collection and analysis, decision to publish, or preparation of the manuscript.

**Competing interests:** The authors have declared that no competing interests exist.

between transcript and protein levels. In fact, model systems including NIH3T3 mouse fibro-blasts, HeLa cells, or human and mouse T cells all showed a correlation coefficient of a mere ~ 0.4 [11–14]. Whether this limited correlation coefficient is a feature of all cells at steady state conditions, and whether it changes during differentiation or upon receiving external cues is to date not known. In particular studies in T cells are lacking.

In addition to transcriptional and metabolic alterations, post-transcriptional and post-translational mechanisms are critical determinants for T cell effector function [15]. For example, in activated Jurkat T cells, as much as 50% of the transcriptome changes was attributed to alterations in mRNA stability [16]. Untranslated regions (UTRs) are key regulatory elements in this process, as they are required for the modulation of mRNA transcript stability, translation initiation, and translation efficiency [17, 18]. Several motifs within the 3'UTR contribute to this regulation. For example, AU-rich elements (AREs) are hubs for RNA binding proteins which in turn determine the fate of mRNA, as exemplified for the key pro-inflammatory cytokines IFN-γ, TNF-α and IL-2 [19–21]. Whether this observed discrepancy of mRNA and protein levels for IFN-γ, TNF-α and IL-2 also holds true for other gene products, and whether gene regulation is defined by specific gene classes, or whether specific features in mRNA influence the protein output in T cells is to date not well understood.

Here, we studied how T cell differentiation and T cell activation modulates the relation between mRNA and protein expression in human CD8+ T cells. We found that independent of the differentiation status the overall correlation coefficient of mRNA and protein in T cells remained unchanged. Nevertheless, the relation of mRNA expression with protein expression was gene class-specific, and it associated with specific sequence characteristics. Gene class specificity even extended to the effect on mRNA and protein levels of AU rich elements.

## Materials and methods

### Data acquisition

For analysis of mRNA and protein abundance during T cell differentiation, we retrieved paired RNA-seq data with paired MS data from *Böttcher et al.* (Supplemental table 3 of reference [22]), and the National Center for Biotechnology Information (NCBI) gene expression omnibus (GEO) repository under the accession no: GSE63144. To remove biological variation between samples, biological replicates of human naïve ($T_N$; CD45RA+ CD62L$^{high}$; n = 3/4; RNA-seq/MS), central memory ($T_{CM}$; CD45RO+ CD62L$^{high}$ CX3CR1-; n = 3/4) and effector-memory ($T_{EM}$; CD45RO+, CD62L$^{low}$, CX3CR1$^{high}$; n = 3/4) CD8+ T cells were averaged per group for downstream analysis. Of note, sample-to-sample variation was limited between replicates [22]. Samples of memory CD45RO+ CD8+ T cells ($T_{MEM}$; n = 12/16) from *Bottcher et al.* were averaged.

Sequencing runs of the same samples were concatenated. For analysis on CD8+ T cell activation, we used our previously published data [23] on the NCBI GEO repository under the accession no: GSE125497. MS data were acquired on the Proteome Xchange Consortium via the PRoteomics IDEntifications Database (PRIDE) repository with the dataset identifier no: PXD012874. For comparative analysis presented here, RNA-seq and MS data from each biological replicate containing IFN-γ producers (n = 3), IL-2 producers (n = 3), IFN-γ/IL-2 double producers (n = 3), and double negative producers (n = 3) were pooled, and the average expression was used.

### RNA-seq analysis

All samples were analyzed with FASTQC (Babraham institute: https://www.bioinformatics.babraham.ac.uk/projects/fastqc/) and passed our quality control (S2A Fig). Briefly, an average

of $10.5 \times 10^6$ (*Böttcher et al.*) and $21.4 \times 10^6$ (*Nicolet et al.*) reads per sample was used for quasi-mapping using Salmon (version 0.13) on the human coding transcriptome hg38-release 92 from ENSEMBL [24]. Normalized counts in 'Transcripts per kilobase per million' (TPM; which corrects for transcript length and library size) was calculated by Salmon, and ENSEMBL annotations were used to filter for protein-coding transcripts, and to sum up the TPMs per genes. Genes with TPM>0.1 (*Bottcher et al.*) and TPM>0.01 (*Nicolet et al.*) were considered as expressed. Due to shallower sequencing depth in samples from *Bottcher et al.*, a higher cut-off was used. TPMs per gene were log10-tranformed and averaged across samples as described above.

## MS analysis

For samples from *Nicolet et al.* [23], the RAW mass spectrometry files were processed as described [23]. The Perseus (version 1.6.0.7) plugin for Protein ruler methodology [25] was used to calculate protein copy number (CN) and protein content per cell (in pg per cell). CN values of all 12 biological replicates were averaged and used to integrate with mRNA data. For samples from [22], raw data were not available. LFQ values provided by the authors were used to compute CN values in Perseus. Values from all samples were averaged to integrate with mRNA data.

## RNA-protein relation in different gene classes

Genes encoding secreted proteins were extracted from https://www.proteinatlas.org/humanproteome/tissue/secretome (downloaded Jan 2018), and manually curated for obvious mis-annotations (histones, membrane, TF, RNA-binding, CD molecules, protein without protein evidence, collagen, nuclear, nucleus, ribosomal, mitochondrial). Similarly, genes and proteins encoding CD molecules were extracted with the corresponding gene and protein names (www.uniprot.org/docs/cdlist.txt; downloaded July 2017). Genes of experimentally validated RNA-binding proteins were obtained from [26–28]. Genes encoding for transcription factors, ribosomal proteins, and TCA-cycle were obtained from https://www.proteinatlas.org/humanproteome/proteinclasses (downloaded Jan. 2019). A-U rich element (ARE) cluster content of 3'UTRs was obtained from ARED-plus [29]. Genes with more than one ARE cluster were annotated as ">1 ARE". GC content of genes (pre-mRNA sequence), orthologous (pre-mRNA) sequence conservation between human and Zebrafish (*Danio rerio*), and 5' and 3'UTR lengths were obtained from Ensembl BioMART (ENSEMBL; [30]). For each sequence parameter (Figs 4 and S2), the data were subdivided into 3 groups based on top 25%, intermediate 50%, and bottom 25%.

## Statistical analysis and data visualisation

The mRNA-protein slope is the slope coefficient defined by the linear model (*lm()* function of R 'stats' package) using the formula '*lm(protein ~ mRNA)*' where protein is the log10(*copy number*) and mRNA is the log10(*TPM*). The mRNA to protein ratio was calculated for each gene class using the average mRNA in log10(*TPM*) plotted against the average protein in log10(copy number). Statistical analysis was performed in R using a two-tailed t-test. The resulting p-values were adjusted using Benjamini-Hochberg procedure. To test for statistical significance of regressions, we used an F-test. To test statistical significance of correlations, we used cor.test() of the R {stats} package. Differences were considered significant if adjusted p-value <0.05, depicted in each graph. Plots were generated with ggplot2 version 3.0, and with GraphPad PRISM version 8. Heatmaps were generated with pHeatmap version 1.0.10. Scripts used

in this study are available on Github at https://github.com/BenNicolet/mRNA_and_protein_intregration_in_CD8_Tcells.

# Results

## Limited correlation of mRNA and protein abundance in CD8$^+$ T cell populations

We first asked how the mRNA and protein expression related to each other in human T cells. For this analysis, we used previously published RNA-sequencing (RNA-seq) and mass spectrometry (MS) data of peripheral blood-derived human CD8$^+$ T cells that were naïve (T$_N$), central memory (T$_{CM}$) or effector-memory (T$_{EM}$) cells [22]. In addition, we included datasets of CD8$^+$ T cells activated for 4h with PMA/Ionomycin (T$_{Activated}$) [23]. We mapped RNA-seq reads and computed normalized read counts in transcript per kilobase per million (TPM), which corrects for transcript length and sequencing depth, a critical feature for accurate quantification of RNA abundance (S1 Fig). Across all T cell populations, we detected 14,195 genes expressed on average, representing ~70% of all protein-coding genes (S2A Fig). The mass spectrometry measurements of paired samples identified 6,069 proteins across all blood-derived populations, and 4,626 proteins in T$_{Activated}$. To estimate protein abundance, we computed the protein copy number (CN) from MS data using the proteomic ruler methodology [25]. mRNA and protein abundance spanned over 7.2 and 6.2 orders of magnitude, respectively (Fig 1A

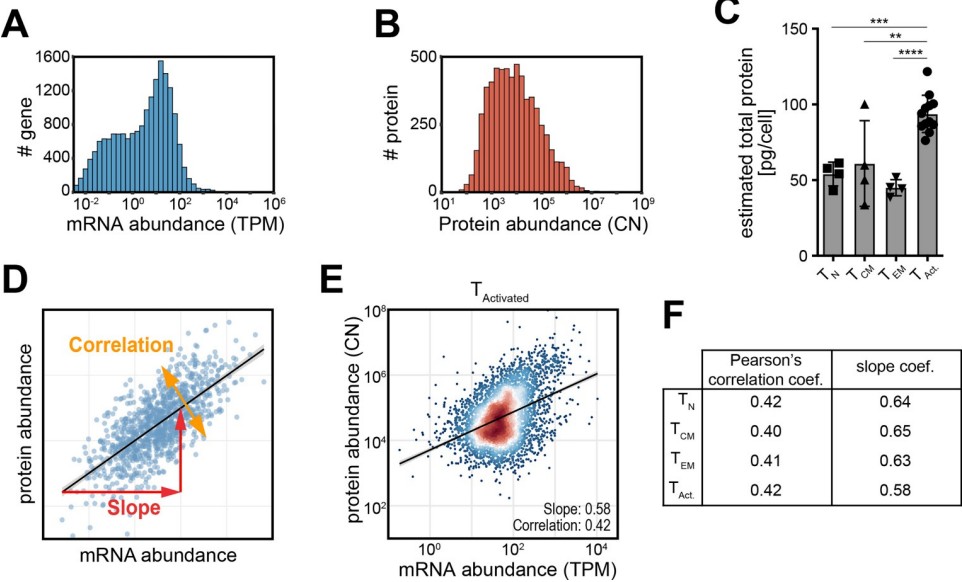

**Fig 1. Transcriptome and proteome integration in human CD8$^+$ T cells.** (**A, B**) mRNA abundance in transcript per kilobase per million (TPM) and protein abundance data in protein copy number (CN) of CD8$^+$ T cells that were activated for 2 days with αCD3/αCD28, cultured for 4 days, and were then re-activated for 4h with PMA-Ionomycin (T$_{Activated}$; n = 12; from [23]). (**C**) Estimated protein mass per cells using the proteomic ruler methodology [25] of blood-derived, naïve (T$_N$; CD45RA$^+$ CD62L$^{high}$; n = 4), central memory (T$_{CM}$; CD45RO$^+$ CD62L$^{high}$ CX3CR1$^-$; n = 4) and effector-memory (T$_{EM}$; CD45RO$^+$, CD62L$^{low}$, CX3CR1$^{high}$; n = 4) CD8$^+$ T cell subsets and for CD8$^+$ T cells from (A). The p-value resulting from two-tailed t-test is indicated in (C). (**D**) Graph representing mRNA and protein correlation (such as Pearson's correlation) and slope of a linear regression. Of note, the correlation coefficient represents a numerical integration of the dispersion of the data points and is not a measure of the angle of the regression line. (**E**) Integrated mRNA and protein abundance in T$_{Activated}$. The Pearson's correlation coefficient, the linear regression (grey line), and its mRNA-to-protein slope (also known as Beta-coefficient) are indicated in the graph. (**F**) Table indicating the Pearson's correlation coefficient and the mRNA-to-protein slope for T cell populations in (C).

and 1B; S1 and S2 Tables). With the proteomic ruler methodology, we found that blood-derived CD8+ T cells contain on average 47.35 pg of protein per cell (n = 12; Fig 1C). In line with previous reports [31, 32], the protein content per cell nearly doubled in activated CD8+ T cells to 93.75 pg/cell (n = 12; Fig 1C). This finding thus indicates a substantial rearrangement of the T cell proteome upon activation.

The use of TPM (transcriptome) and CN (proteome) units allowed us to integrate mRNA expression levels with protein measurements. 6,013 gene products were detected both at mRNA and protein levels in blood-derived T cells, representing ~99,1% of detected protein and ~42% of all genes detected at mRNA level. Activated T cells contained 4,583 gene products with co-detected mRNA and protein levels. Intriguingly, irrespective of T cells differentiation or activation status, the mRNA-protein correlation was almost identical (Pearson's coefficient = 0.40–0.42; Spearman's coefficient: 0.36–0.37; Fig 1D–1F, p-value: <2e-16). We also calculated the overall mRNA-to-protein slope, which reflects how mRNA abundance translates into protein abundance (Fig 1D). The mRNA-to-protein slope, visualized by the regression line, was 0.58–0.65 (Fig 1E and 1F, p-value <2e-16). Notably, even though the mass per cell nearly doubled in activated T cells (Fig 1C), this had no effect on the correlation and slope between mRNA and protein (Fig 1E and 1F). In conclusion, the overall abundance of mRNA in human CD8+ T cells only moderately correlates with protein expression and is independent of the cell status.

## Distinct gene functions show differential correlation of mRNA with protein expression

We next questioned whether this limited correlation of mRNA expression with protein output was equal for all genes, or whether different gene classes displayed specific expression patterns. We first examined the relation between mRNA and protein abundance in activated CD8+ T cells. We specifically isolated secreted proteins, CD molecules, transcription factors, RNA-binding proteins (RBPs), ribosomal proteins and Tricarboxylic (citric) acid cycle gene products (TCA) (S1 and S2 Tables). Of note, because protein secretion was blocked by Monensin during activation (which did not alter expression levels of effector molecules [33]), we could also measure the expression of 22 secreted proteins. Secreted proteins and the 1174 RBPs displayed a Pearson's correlation coefficient of 0.57 and 0.51, respectively, which was above average of all genes combined (~0.42; Figs 1E and 2A), and were only outcompeted by the 24 genes involved in TCA cycle (0.7) and the 93 mammalian ribosomal proteins (0.72). The 84 CD molecules were close to the average with a Pearson's correlation coefficient of 0.43. In contrast, the 176 transcription factors and 66 mitochondrial ribosomal proteins showed a correlation coefficient of only 0.31 and 0.03, respectively (Fig 2A). These finding highlights that subsets of genes, defined by their function, have a distinct mRNA-protein correlation in activated CD8+ T cells.

We next examined whether the differential mRNA-protein correlation for these gene classes was also observed throughout T cell differentiation. Mitochondrial ribosomal proteins displayed with -0.20 to -0.26 even a negative Pearson's correlation coefficient, indicating that as mRNA levels increase, lower protein levels are observed (Fig 2B). Conversely, mammalian ribosomal proteins, TCA cycle genes, RBPs and CD molecules remained their Pearson's correlation coefficient during T cell differentiation (Fig 2B). Nevertheless, T cell activation resulted in an increased coefficient for TCA cycle genes and RBPs, and a decreased coefficient for CD molecules (Fig 2B). For secreted proteins, the Pearson's correlation coefficient increased through differentiation and upon T cell activation (Fig 2B). Similar patterns were observed with Spearman's correlation coefficient and with the mRNA-to-protein slope for gene classes (Figs 2C and S2B). Thus, mRNA and protein levels correlate in a gene class-specific manner, which only alters slightly through T cell differentiation and upon activation.

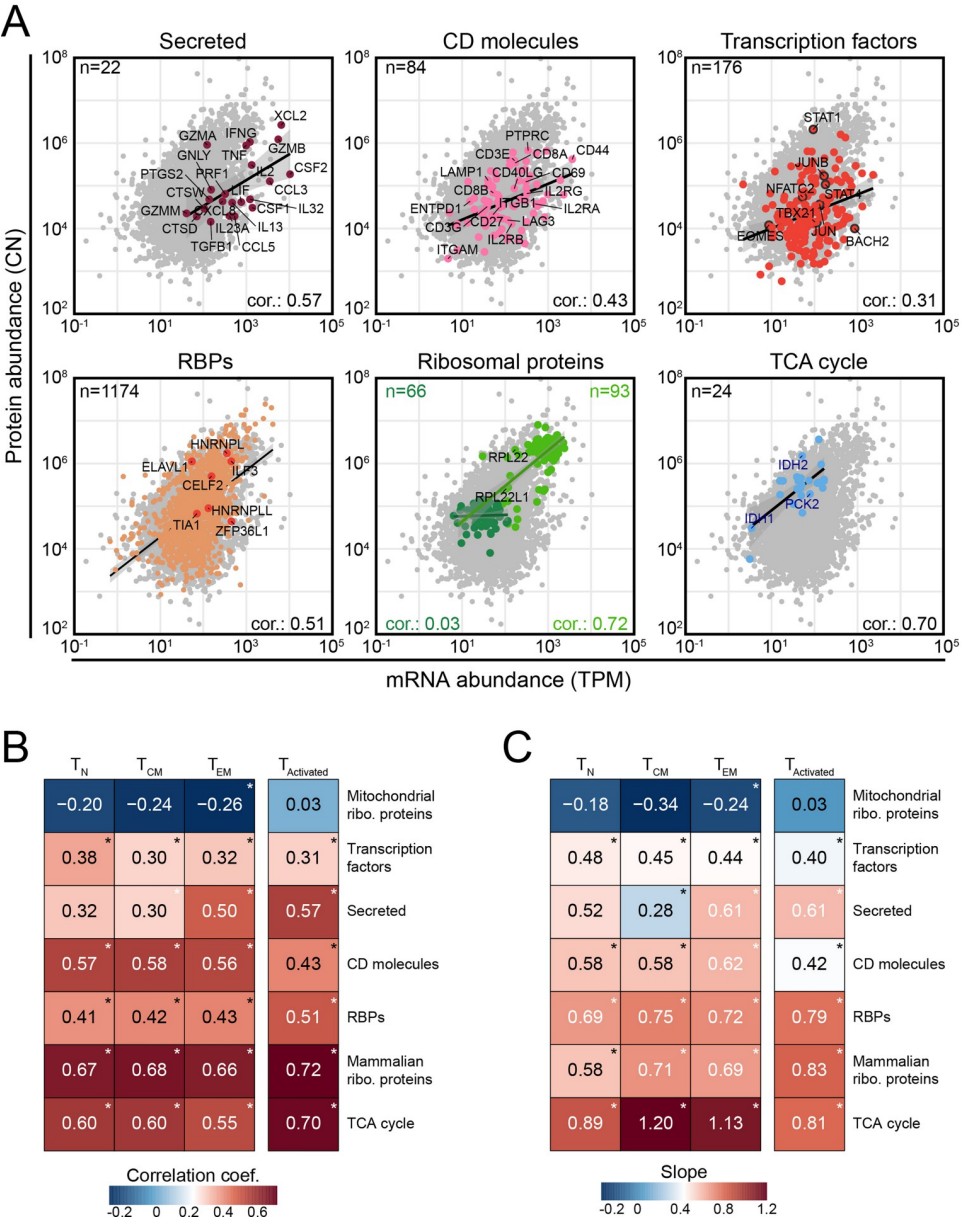

**Fig 2. Gene class-specific mRNA and protein abundance.** (**A**) Integrated mRNA (TPM) and protein abundance (CN) of secreted proteins, CD molecules, transcription factors, RNA-binding proteins (RBPs), mammalian (light green) and mitochondrial (dark green) ribosomal proteins, and TCA (citric acid) cycle proteins of activated CD8+ T cells ($T_{Activated}$). The number of genes per class, the Pearson's correlation coefficient, and regression line (linear model) were calculated for each class and are indicated in the panels. The correlation was performed independently for mammalian (light green) and mitochondrial (dark green) ribosomal proteins. (**B, C**) Pearson's correlation coefficient (B) and mRNA-to-protein slope (also known as Beta-coefficient) (C) of the integrated mRNA and protein abundance of gene classes indicated in (A) during T cell differentiation and upon T cell activation. "*" indicates correlation and regression with p-value <0.05.

## Gene classes display distinct gene characteristics

Differential gene characteristics could explain the diversity of mRNA and protein relation in gene classes. We therefore examined the sequence conservation of the different gene classes, by comparing the sequence homology at the gene level between human and zebrafish (*Danio*

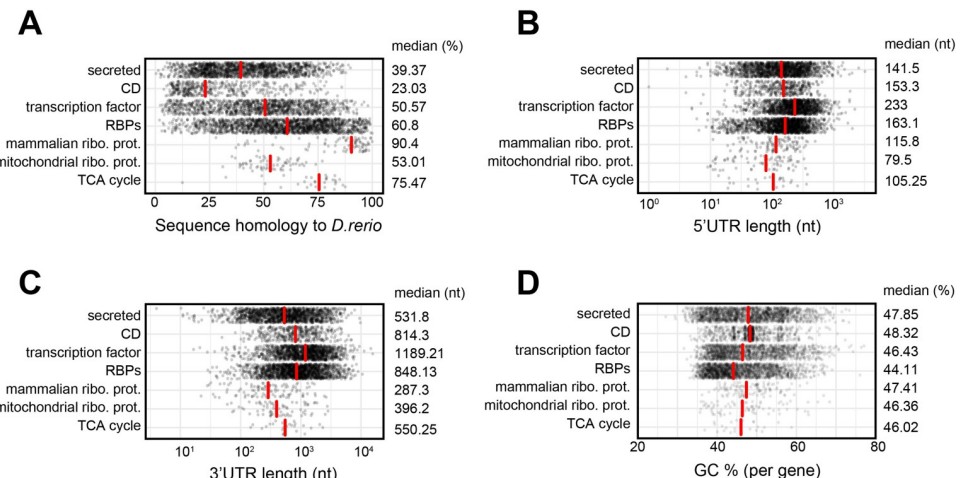

**Fig 3. Distinct gene characteristics are associated with gene functions.** (**A-D**) Relation between gene classes (Fig 2) and sequence homology of human genes with Zebrafish (*D. rerio*) (A), the length of the 5'UTR (B) and 3'UTR (C), and the GC content (D). Red bars indicate the median values per gene class, which are also indicated numerically at each respective graph. nt: nucleotides.

*rerio*). In particular the sequence of mammalian ribosomal proteins was highly conserved (median: 90.4% sequence homology), followed by TCA cycle genes with a median value of 75.47% sequence homology (Fig 3A). Mitochondrial ribosomal proteins displayed only a median sequence homology of 53.01%, comparable to transcription factors with 50.57% and RBPs with 60.8%, with the latter two displaying a wide spread in sequence conservation (Fig 3A). CD molecules and secreted proteins had the lowest sequence conservation with 23.03% and 39.37% homology, respectively (Fig 3A).

Untranslated regions of RNAs are critical components in defining the RNA levels and the protein output of a given mRNA, and the length of 5'UTR and 3'UTRs is contributing [34]. We therefore also compared the length of 5'UTRs and 3'UTRs across gene classes, and we found that they show similar patterns (Fig 3B and 3C). For example, ribosomal proteins and TCA cycle genes tended to have shorter UTRs than other gene classes (Fig 3B and 3C). In contrast, transcription factors had longer UTRs than other gene classes (medians: 5'UTR: 233nt; 3'UTR: 1189.21nt), followed by RBPs (5'UTR: 163.1nt; 3'UTR: 848.13nt). Secreted protein and CD molecules, had similar length distributions, and their UTR length spanned over 2 orders of magnitude for 5'UTR length and 3 orders of magnitude for 3'UTR length (Fig 3B and 3C).

Also the GC content can influence a gene's expression. Even though the GC content of all genes ranged from 25% to 77%, we did not observe large differences in medians between gene classes. Nonetheless, RBPs had a somewhat reduced median GC% and CD molecules a somewhat increased median GC% (Fig 3D). Combined, different levels of sequence homology and of UTR length, in particular, is observed across gene classes.

## Gene and transcript characteristics associate with mRNA and protein abundance

We next investigated how gene characteristics related to expression levels. We used mRNA and protein abundance of activated CD8[+] T cells, and examined the relation between expression and sequence conservation (Fig 4A). Of the 4,583 gene products that were co-detected at mRNA and protein level in PMA-Ionomycin activated CD8[+] T cells, 4,568 (99.7%) gene products were present in both human and zebra fish (*Danio rerio*). We divided these genes high

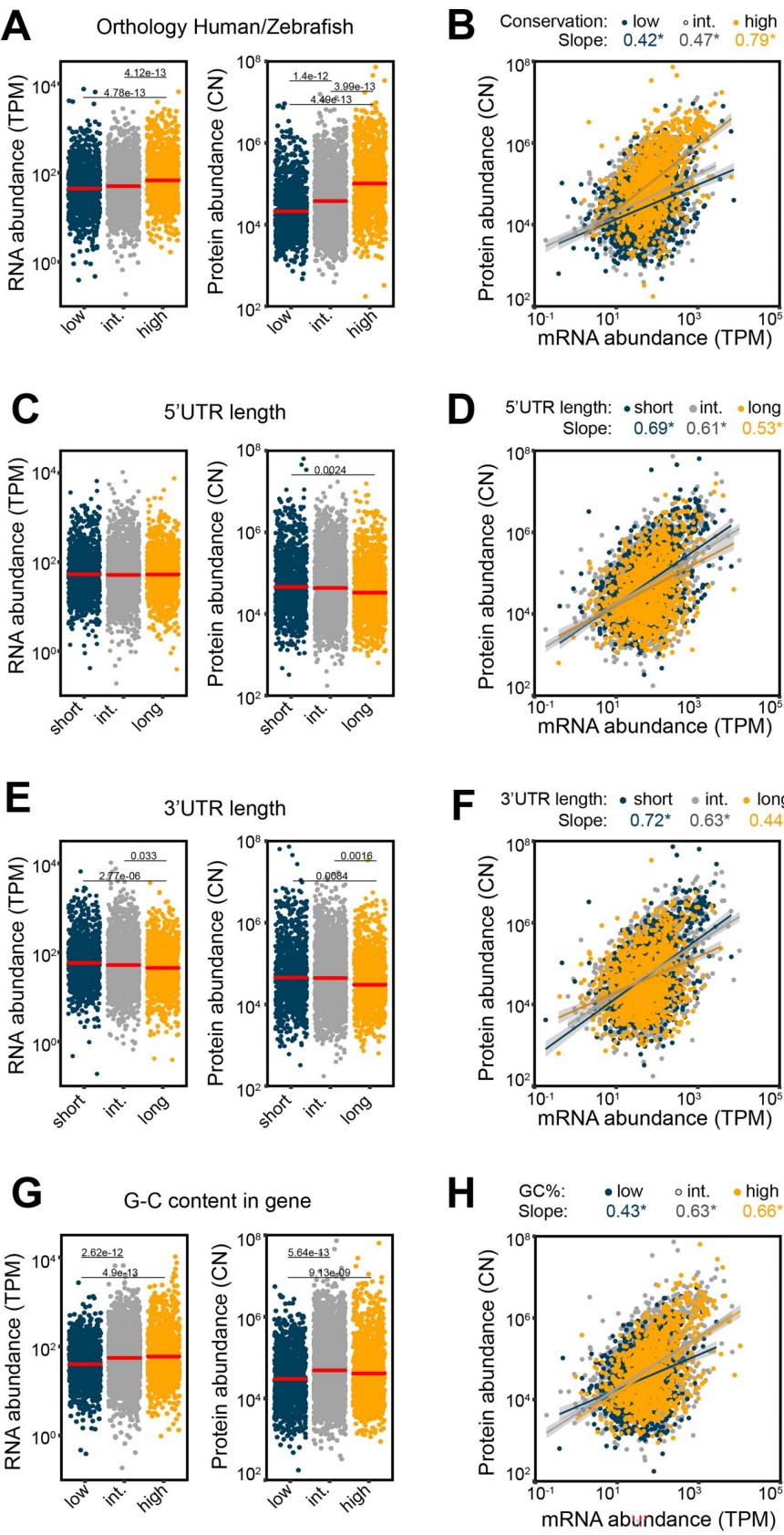

**Fig 4. Altered mRNA and protein abundance is associated with gene characteristics.** (**A**, **B**) The effect of sequence conservation on mRNA (A, left panel) and protein abundance (A, right panel) and the relation thereof (B) in PMA-Ionomycin activated CD8$^+$ T cells. Genes products were stratified according to S3 Fig, in: (**A, B**) high (>76.9%), intermediate (between 50–76.9%) and low (<50%) orthologous sequence conservation between Human and Zebra-fish (*Danio rerio*); (**C, D**) short (<~50nt), intermediate (~50nt- ~200nt), and long 5'UTRs (>~200nt); (**E, F**) short (<~100nt), intermediate (~100- ~1,000nt), and long (>~1,000nt) 3'UTRs; (**G, H**) high (>50.9%), intermediate (39.7%<GC<50.9%) and low (<39.7%) GC content per gene. The slope of the regression line of each group is color-coded in B, D, F, H. Differences between groups were assessed with a two-tailed t-test followed by adjustment of p-value using the Benjamini-Hochberg procedure. Adjusted p-value are indicated. Only gene products that are detected both at mRNA and protein levels are shown. The mean per group (red line) is indicated in A,C,E,G. TPM: Transcripts per kilo-base per million; CN: Protein copy number. "*" indicates correlation and regression with p-value <0.05.

sequence homology (top 25% of genes: >76.9% conservation), intermediate (25–75% of genes: 50%-76.9% conservation), and low sequence homology (bottom 25% of genes: <50% conservation; Figs 4A and S3). High conservation correlated with the highest mRNA abundance and protein abundance in activated T cells (Fig 4A). This translated in a high mRNA-protein slope (0.79) for these well conserved transcripts, while intermediate and low conserved genes reached only 0.47 and 0.42, respectively (Fig 4B; all regressions: p-value <2.2e-16). Thus, highly conserved genes display the highest expression in T cells.

We then turned to the relation between UTR lengths and gene expression, which we stratified into genes with short (shortest 25%: < ~50nt), intermediate (25–75%), or long 5'UTRs (longest 25%: > ~200nt; Figs 4C and S3), and short (bottom 25%, < ~100nt), intermediate (25–75%), and long (top 25%, > ~1,000nt) 3'UTRs (Figs 4E and S3). mRNA abundance was not affected by the length of the 5'UTR, and only slightly reduced by long 3'UTRs (Fig 4C and 4E). A short 5'UTR resulted in slight but significantly higher protein levels (Fig 4C), as previously reported in yeast (*Saccharomyces cerevisiae*) [35]. A similar correlation of long 3'UTRs with reduced protein expression was observed (Fig 4E). This also translated into higher mRNA-protein slopes for genes with long 5'UTRs, and even more so for genes with long 3'UTRs (Fig 4D and 4F; all regressions: p-value <2.2e-16). Thus, the length of the UTRs—and in particular that of the 3'UTR—influences the mRNA and protein level in activated T cells.

Genes with a high GC content (top 25% of genes: GC>50.9%) had a significantly higher mRNA and protein abundance compared to genes with low (GC<39.7%,) or intermediate GC content (39.7%<GC<50.9%), which was also reflected in the RNA-protein slope (Fig 4G, 4H and S3; all regressions: p-value <2.2e-16). Thus, gene characteristics not only associate with particular gene functions, but also associate with altered mRNA and protein expression in CD8$^+$ T cells.

## AU-rich elements influence the relation between mRNA and protein

The influence of 3'UTR on gene expression is not limited to length. In fact, 3'UTR contain many binding sites for RNA-binding proteins and microRNAs. A key regulatory element in 3'UTRs are A-U rich elements (AREs) [17, 19, 36]. Indeed, ARE-containing transcripts such as *IFNG*, *TNF* and *IL2*, are strictly regulated by ARE-binding RBP especially during T cell activation when these cytokines are secreted [15]. To define whether AREs associate with mRNA and protein abundance in CD8$^+$ T cells, we used ARED-plus AU-rich element annotations from [29]. ARED-plus annotations define 1 cluster as `WWWW (AUUUA) WWWW` (where W is A or U), 2 clusters as `AUUUAUUUAU` or `UAUUUAUUUA` flanked by W or WW, and 3 and over clusters are defined as `AUUUAUUUAUUUA` and additional repeats of AUUUA [29]. When we integrated the number of ARE clusters with mRNA abundance in T$_{MEM}$ cells, mRNAs containing one or more ARE clusters in 3'UTR expressed higher mRNA levels (Fig 5A).

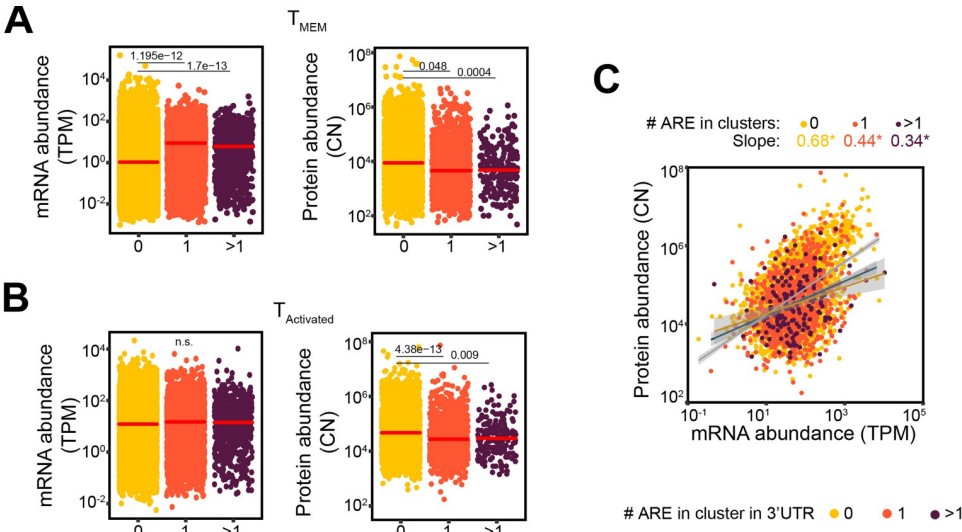

**Fig 5. ARE clusters differentially modulate mRNA and protein levels in resting and activated CD8+ T cells. (A-B)**
mRNA (TPM) and protein (CN) abundance in (A) CD45RO+ CD8+ memory T cell subsets (T$_{MEM}$; n = 16; from [22])
and (B) T$_{Activated}$ CD8+ T cells of all genes, grouped according to the number of AU-rich element in cluster within the
transcript's 3'UTR clusters according to ARED-plus annotation (A; see methods). Differences between groups were
assessed with a two-tailed t-test followed by adjustment of p-value using the Benjamini-Hochberg procedure. Adjusted
p-value are indicated. (C) and integrated mRNA and protein abundance. The slope of the regression line of each group
is color-coded in (A, B), and only gene products that are detected both at mRNA and protein levels are shown. TPM:
Transcripts per kilo-base per million; CN: Protein copy number. n.s.: not significant. "*" indicates correlation and
regression with p-value <0.05.

In activated T cells, we did not observe differences on mRNA levels in the presence or
absence of ARE clusters (Fig 5B), indicating that AREs may be more important in defining
transcript expression in non-activated T$_{MEM}$ cells than in activated T cells. Nonetheless, ARE
clusters within the 3'UTR significantly affected the protein abundance both T$_{MEM}$ cells and
activated T cells (Fig 5A and 5B), indicating that AREs consistently modulate the protein out-
put irrespective of the activation status of T cells. This is consistent with the notion that AREs
can modulate both mRNA stability and mRNA translation [37]. Consequently, ARE-contain-
ing transcripts displayed a lower mRNA-protein slope with 0.34–0.44 than transcripts without
ARE clusters (0.68; Fig 5C; 0 and 1ARE: p-value <2.2e-16, >1ARE: p-value 2.15e-06). Hence,
the presence of ARE clusters within 3'UTRs modifies the mRNA and protein relation in acti-
vated T cells.

## AU-rich elements mostly affect secreted proteins

We next examined the effect of ARE clusters on mRNA expression and protein output in gene
classes. ARE is thought to drive expression of transcripts that require a rapid on-off produc-
tion, such as certain CD molecules, transcription factors and secreted proteins [15]. For CD
molecules and transcription factors, we did not detect a role of ARE clusters on mRNA or pro-
tein expression levels in T$_{MEM}$, or in activated T cells (S4 Fig). In addition, CD molecules lack-
ing ARE-Clusters were subject to strong differences in mRNA-protein slopes between T$_{MEM}$
and activated T cells (0.56 to 0.36; respective p-values: 7.079e-08, 0.0017), while genes with
1ARE had marginal changes (0.68 to 0.53; respective p-values: 0.00015, 0.0079), pointing to
ARE-independent effects on gene expression for these genes at activation (Fig 6A and 6B).
Transcription factors only showed marginal if any changes in the mRNA-protein slopes when
comparing genes containing 0 or 1ARE cluster, in T$_{MEM}$ with activated T cells (Fig 6C and 6D).

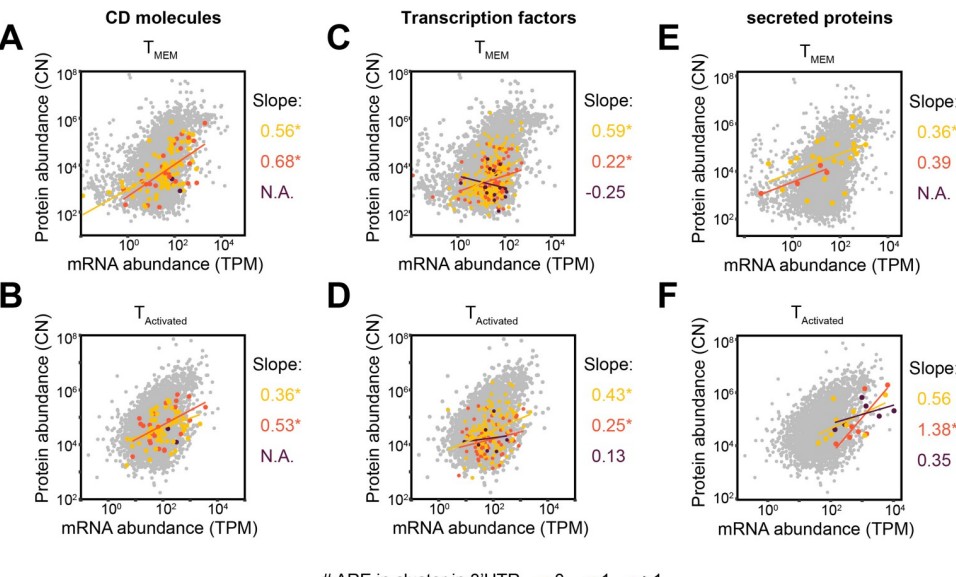

**Fig 6. ARE cluster content differentially affect gene classes.** (**A-F**) Integrated mRNA (TPM) and protein (CN) abundance of genes co-detected at mRNA and protein levels, for (**A, B**) CD molecules, (**C, D**) transcription factors, or (**E, F**) secreted proteins. Genes were stratified according to the number of AU-rich element in cluster within the transcript's 3'UTR, for (**A, C, E**) $T_{MEM}$ or (**B, D, F**) $T_{Activated}$ CD8$^+$ T cells. The linear regression and its slope are color-coded for each group. Difference between groups were assessed with a two-tailed t-test followed by adjustment of p-value using the Benjamini-Hochberg procedure. Adjusted p-value are indicated. TPM: Transcripts per kilo-base per million; CN: Protein copy number. "*" indicates correlation and regression with p-value <0.05.

In contrast, the slope of genes containing >1ARE cluster increased from -0.25 (p-value: 0.46) in $T_{MEM}$ to 0.13 (p-value: 0.98) in activated T cells. Thus, even though the mRNA-protein slope changed, it remains below that of transcription factors containing 0 or 1ARE cluster. Of note, due to the low number of mRNA-protein pairs, these finding should be considered as mere indication for the regulatory effect of ARE clusters.

Secreted proteins did not display alterations in mRNA abundance in $T_{MEM}$ CD8$^+$ T cells (S4E and S4F Fig). Yet, the average mRNA levels went down with increasing numbers of ARE clusters (means: 0ARE: 1.2 TPM; 1ARE: 0.38 TPM; >1ARE: 0.14 TPM; S4E and S4F Fig), which appeared specific for secreted molecules (S4 Fig). Interestingly, even though T cell activation resulted in overall increased mRNA levels, this was most apparent for mRNAs with >1ARE cluster (>1ARE: mean 30.4 TPM; S4E and S4F Fig). None of the 23 mRNAs containing >1ARE clusters in $T_{MEM}$ CD8$^+$ T cells had detectable protein levels (Figs 6E and S4E). Upon T cell activation, 6 out of the 17 mRNAs with >1ARE clusters showed protein production (CCL3, CSF2, CXCL8, IL2, PTGS2, TNF) in activated T cells (Figs 6F and S4F). The mRNA-to-protein slope for secreted proteins of transcripts in $T_{MEM}$ with 0 or 1ARE cluster was similar with 0.36 and 0.39, respectively (Fig 6E; respective p-values: 0.015, 0.068). However, in activated T cells, the mRNA-protein slope increased irrespective of the ARE cluster content (Fig 6F). Only 7 and 6 mRNA/proteins pairs were co-detected in 1 and >1ARE clusters, respectively (1ARE: CSF1, IFNG, IL13, IL23A, LIF, TGFB1, XCL2; >1ARE: CCL3, CSF2, CXCL8, IL2, PTGS2, TNF), yet the slope differed between these two groups (1ARE: 1.38 (p-value: 0.023); >1ARE: 0.35 (p-value: 0.14); Fig 6F). Again, due to the low number of mRNA-protein pairs these findings should only be considered indicative. Together, our findings indicate that ARE clusters associate with suppression of protein abundance in secreted proteins and transcription factors, but not in CD molecules.

## Discussion

With an integrated analysis of transcriptome and proteome data from primary human CD8[+] T cells, we show here that the mRNA expression only moderately correlates with protein expression, a feature that is independent of T cell differentiation or activation. Even though we did not detect peptides for all transcripts by MS analysis, the limited correlation holds true at least for the ~40% of the protein-encoding mRNAs of which we used the protein expression data.

The limited correlation in mRNA and protein abundance was previously reported in other model systems [11–14], and has been attributed to differences in genetic variation, transcriptional rates, mRNA stability for mRNA abundance, and to protein synthesis and degradation rates for protein abundance [38, 39]. Which of these parameters is the prime driver of the mRNA expression levels and protein output may be gene-specific and may depend on the differentiation and activation status of a cell. Indeed, whereas the overall correlation between mRNA and protein was limited, we found gene class-specific mRNA and protein expression patterns. Mitochondrial ribosomal proteins are only stable when bound to the mitochondrial ribosome [40]. In contrast, the mRNA of mammalian ribosomal proteins are considered stable [41], and mammalian ribosomal proteins are known to be expressed in ribosome-unbound form as well [42]. Differential degradation of unbound mitochondrial ribosomal proteins and mammalian ribosomal protein could thus lead to the observed discrepancy in mRNA and protein abundance for both protein classes. Similarly, TCA genes did not only express high levels of mRNA and protein but also showed a strong correlation of mRNA abundance with protein abundance. It is therefore conceivable that the conservation of these so-called 'housekeeping' genes resulted in high transcription and translation efficiency [43].

Protein stability is another important determinant of protein expression levels [12, 44]. Mammalian ribosomal proteins and nucleosomes are more stable than secreted proteins, as are zinc-finger containing proteins (both RNA-binding proteins and transcription factors [44, 45]). However, protein stability is also cell-type specific [46]. Although we did not directly address protein stability in differentiating and activated human CD8[+] T cells, the high correlation for mRNA and protein expression for TCA proteins and ribosomal protein match with these studies and reiterate the positive relation of gene sequence conservation with mRNA and protein expression levels. Evolutionary sequence conservation can also result in altered GC-content of the mRNA coding region, which translates into higher mRNA levels and translation efficiency [47, 48]. This could explain why high GC content associates with higher mRNA and protein levels. Complementary to our findings, GC content manipulation of the IL6 and IL2 gene resulted in higher mRNA stability, ribosome occupancy and protein output [47, 49]. GC content also influences codon usage and secondary structures, which in turn increases translation efficiency and protein output [50–53].

Not only sequence conservation, but also the length of 5' and 3'UTR influences the mRNA and protein abundance. UTR length alone, however, is unlikely to be the sole factor that influences mRNA and protein abundance. Similar to the increased presence of ARE clusters in longer 3'UTRs [34], the overall likelihood of containing regulatory motifs and structures increases with length. Notably, murine T cells activated for 48h with aCD3/CD28 beads display substantial shortening of 3'UTRs [54]. Whether this also occurs after 4h PMA-Ionomycin activation is yet to be determined. It is however conceivable that alterations in 3'UTR length can contribute to mRNA expression in activated T cells by altering the content of miRNA seeds and thus mRNA stability [55].

Also AU-rich elements strongly influence the mRNA and protein levels [20, 37, 56]. Interestingly, ARE-mediated regulation in CD molecules did not alter mRNA and protein expression. Only ARE-containing transcripts encoding secreted proteins and transcription factors

had high mRNA levels while having low to undetectable protein levels in memory T cells. Upon activation, however, these transcript showed high mRNA and high protein levels, reflecting the discrepancy of mRNA and protein levels of IFN-γ in memory and tumor-infiltrating T cells due to a ARE-dependent translational block [37, 56]. What makes transcripts susceptible to this regulation is yet to be determined, but it could possibly stem from subtle differences in position and spacing between AREs, the surrounding RNA structures, or the interplay with other regulatory mechanisms on the same transcript.

In summary, we show that the limited relation between the mRNA and protein levels is gene class specific and is defined by sequence conservation and by sequences and structures of UTRs. Our findings serve as valuable resource to further study the relation between sequence characteristic and gene expression in CD8$^+$ T cells.

## Supporting information

**S1 Fig. Analysis workflow of RNA sequencing and Mass spectrometry data.** Workflow of the analysis of RNA-sequencing (RNA-seq) and Mass spectrometry (MS) data of blood-derived naïve (T$_N$; CD45RA$^+$ CD62L$^{high}$; n = 3), central memory (T$_{CM}$; CD45RO$^+$ CD62L$^{high}$ CX3CR1$^-$; n = 3), effector-memory (T$_{EM}$; CD45RO$^+$, CD62L$^{low}$, CX3CR1$^{high}$; n = 4) and memory CD45RO$^+$ CD8$^+$ T cells (T$_{MEM}$; n = 16) CD8$^+$ T cell subsets and of CD8$^+$ T cells that were activated for 2 days with αCD3/αCD28, cultured for 4 days, and were then re-activated for 4h with PMA-Ionomycin (T$_{Activated}$; n = 12; from [23]).
(JPG)

**S2 Fig. Quality control of RNA sequencing and Mass spectrometry data.** (**A**) Quality control (QC) using FASTQC, number of reads mapped to transcriptome, number of gene expressed and protein detected for blood-derived, naïve (T$_N$; CD45RA$^+$ CD62L$^{high}$; n = 3), central memory (T$_{CM}$; CD45RO$^+$ CD62L$^{high}$ CX3CR1$^-$; n = 3) and effector-memory (T$_{EM}$; CD45RO$^+$, CD62L$^{low}$, CX3CR1$^{high}$; n = 4) CD8$^+$ T cell subsets and for CD8$^+$ T cells that were activated for 2 days with αCD3/αCD28, cultured for 4 days, and were then re-activated for 4h with PMA-Ionomycin (T$_{Activated}$; n = 12; from [23]). (**B**) Spearman's correlation of the integrated mRNA and protein abundance of gene classes indicated in Fig 2 during T cell differentiation and upon T cell activation.
(JPG)

**S3 Fig. mRNA and/or protein abundance associates with gene characteristics.** Distribution and color-coded cut-off (top 25%; mid. 50%; bottom 25% of all genes) for stratifying orthologous gene sequence conservation between Human and Zebrafish (in %), GC content (in %), 5' and 3'UTR length (in nucleotides; nt). Only gene products detected at mRNA and protein levels are shown.
(JPG)

**S4 Fig. ARE-mediated regulation of transcripts encoding CD molecules and transcription factors do not alter upon T cell activation.** (**A-F**) mRNA (TPM) and protein (CN) abundance in log10 space for CD molecules (A-B), transcription factors (C-D), or secreted molecules (E-F) in T$_{MEM}$ (A, C, E) or T$_{Activated}$ CD8$^+$ T cells (B, D, F). Genes products are color-coded according to the number of ARE clusters (ARED annotation) within the 3'UTR. Differences were assessed with a two-tailed t-test followed by adjustment of p-value using the Benjamini-Hochberg procedure. N.D.: not detected.
(JPG)

**S1 Table. Transcriptome and proteome data in TActivated.**
(XLSX)

**S2 Table. Transcriptome and proteome data of TN, TCM, TEM and TMEM.**
(XLSX)

## Acknowledgments

We would like to thank Branka Popovic and Nordin Zandhuis for critical reading of this manuscript.

## Author Contributions

**Conceptualization:** Benoît P. Nicolet, Monika C. Wolkers.

**Data curation:** Benoît P. Nicolet.

**Formal analysis:** Benoît P. Nicolet.

**Funding acquisition:** Monika C. Wolkers.

**Investigation:** Benoît P. Nicolet.

**Methodology:** Benoît P. Nicolet.

**Project administration:** Monika C. Wolkers.

**Resources:** Monika C. Wolkers.

**Supervision:** Monika C. Wolkers.

**Visualization:** Benoît P. Nicolet.

**Writing – original draft:** Benoît P. Nicolet, Monika C. Wolkers.

**Writing – review & editing:** Benoît P. Nicolet, Monika C. Wolkers.

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
