## [Decision Letter · Decision Letter 0]

14 Sep 2022

PONE-D-22-19814The relationship of mRNA with protein expression in CD8+ T cells associates with gene class and gene characteristicsPLOS ONE

Dear Dr. Wolkers,

Thank you for submitting your manuscript to PLOS ONE. After careful consideration, we feel that it has merit but does not fully meet PLOS ONE’s publication criteria as it currently stands. Therefore, we invite you to submit a revised version of the manuscript that addresses the points raised during the review process.

Importantly, no new experiments are requested, however you will have to make changes to the manuscript text to address the reviewers' concerns.

We look forward to receiving your revised manuscript.

Kind regards,

Alexander F. Palazzo, Ph.D.

Academic Editor

PLOS ONE

Journal Requirements:

"This research was supported by the Oncode Institute, and the European research council (ERC-Printers 817533)."

"Funded study

NO - The funders had no role in study design, data collection and analysis, decision to publish, or preparation of the manuscript"

5. Please include a copy of Supplemental table 3 which you refer to in your text on page 5.

Reviewers' comments:

Reviewer's Responses to Questions

**Comments to the Author**

1. Is the manuscript technically sound, and do the data support the conclusions?

Reviewer #1: Yes

Reviewer #2: Yes

2. Has the statistical analysis been performed appropriately and rigorously? 

Reviewer #1: Yes

Reviewer #2: No

3. Have the authors made all data underlying the findings in their manuscript fully available?

Reviewer #1: Yes

Reviewer #2: Yes

4. Is the manuscript presented in an intelligible fashion and written in standard English?

Reviewer #1: Yes

Reviewer #2: Yes

5. Review Comments to the Author

Reviewer #1: In “The relationship of mRNA with protein expression in CD8+ T cells associates with gene class and gene characteristics”, Nicolet et. al performed bioinformatic analyses of previously published RNA and protein data from primary human CD8+ T cells. They demonstrate, as has been reported in other contexts, poor correlation between RNA and proteins levels. They furthered their analyses by demonstrating unique correlation patterns between different classes of genes. Overall, this study is scientifically sound, and despite being almost purely in silico, it represents useful data that should be published and available to the field. My concerns are relatively minor, mostly aimed at ensuring clarity in the results:

1) The activated cells were treated with Monensin. Can the authors present data or citations as to whether Monensin impacts protein and/or RNA levels?

2) In figure 4 the authors examine RNA/protein levels when stratifying gene based on various characteristics, such as conservation or UTR length. In Fig 3 they show that genes within each class can have wide ranges for these values. If the authors performed a similar analysis as Fig 4, except grouping the data by gene class, are the trends reported in Figure 4 consistent between each class?

Reviewer #2: The authors have not thought much on the appropriate statistical analysis/test to run, in particular with the regression analysis. Data could have been transformed and presented better. For example, using box plots. See the attached document for comprehensive comments on the manuscript.

6. PLOS authors have the option to publish the peer review history of their article (what does this mean?). If published, this will include your full peer review and any attached files.

Reviewer #1: No

Reviewer #2: No

---

## [Author Response · Author response to Decision Letter 0]

24 Sep 2022

please see point-to-point response uploaded as a document

---

## [Editor Report · Decision Letter 1]

5 Oct 2022

The relationship of mRNA with protein expression in CD8+ T cells associates with gene class and gene characteristics

PONE-D-22-19814R1

Dear Dr. Wolkers,

We’re pleased to inform you that your manuscript has been judged scientifically suitable for publication and will be formally accepted for publication once it meets all outstanding technical requirements.

Kind regards,

Alexander F. Palazzo, Ph.D.

Academic Editor

PLOS ONE
---

## [Editor Report · Acceptance letter]

10 Oct 2022

PONE-D-22-19814R1 

The relationship of mRNA with protein expression in CD8+ T cells associates with gene class and gene characteristics 

Dear Dr. Wolkers:

I'm pleased to inform you that your manuscript has been deemed suitable for publication in PLOS ONE. Congratulations! Your manuscript is now with our production department. 

Kind regards, 

on behalf of

Dr. Alexander F. Palazzo 

Academic Editor

PLOS ONE